# *Enterobacter asburiae* E7, a Novel Potential Probiotic, Enhances Resistance to *Aeromonas veronii* Infection via Stimulating the Immune Response in Common Carp (*Cyprinus carpio*)

Jing Li,[a] Zhao Zhang,[a] Zhi-Bin Wu,[a] Shen-Ye Qu,[a] Gao-Xue Wang,[a] Dong-Dong Wei,[b] Peng-Fei Li,[b] Fei Ling[a]

[a]College of Animal Science and Technology, Northwest A&F University, Yangling, China
[b]Guangxi Key Laboratory of Aquatic Biotechnology and Modern Ecological Aquaculture, Guangxi Academy of Marine Sciences, Guangxi Academy of Sciences, Nanning, Guangxi, China

**ABSTRACT** Probiotics are an alternative strategy for antibiotics, but most probiotics are Gram-positive bacteria suitable for terrestrial animals. Therefore, it is imperative to develop dedicated probiotics for the common carp industry to be ecologically efficient and environmentally friendly. A novel *Enterobacter asburiae* named E7 was isolated from the intestine of healthy common carp and displayed an extensive antibacterial spectrum against *Aeromonas hydrophila, A. veronii, A. caviae, A. media, A. jandaei, A. enteropelogenes, A. schubertii, A. salmonicida, Pseudomonas aeruginosa, Ps. putida, Plesiomonas shigelloides*, and *Shewanella*. E7 was nonpathogenic to the host and susceptible to the majority of antibiotics used in human clinical practice. E7 could grow between 10 and 45°C and between pH 4 and 7 and was extremely resistant to 4% (wt/vol) bile salts. Diets were supplemented with $1 \times 10^7$ CFU/g *E. asburiae* E7 for 28 days. No significant difference in the growth of fish was observed. Expression of immune-related genes *IL-10, IL-8*, and *lysozyme* in common carp kidney was significantly upregulated at weeks 1, 2, and 4 ($P < 0.01$). A significant upregulation of *IL-1β, IFN*, and *TNF-α* expression was observed after week 4 ($P < 0.01$). There was a significant increase in mRNA expression of *TGF-β* at week 3 ($P < 0.01$). Following challenge by *Aeromonas veronii*, the survival rate (91.05%) was significantly higher than observed in the controls (54%; $P < 0.01$). Collectively, *E. asburiae* E7 is a promising new Gram-negative probiotic that can enhance health and bacterial resistance of aquatic animals and could thus be developed as an exclusive aquatic probiotic.

**IMPORTANCE** In the present study, we evaluated for the first time the efficiency of *Enterobacter asburiae* as a prospective probiotic for aquaculture applications. The E7 strain showed extensive resistance to *Aeromonas*, no pathogenicity to the host, and stronger environmental tolerance. We observed that the resistance of common carp to *A. veronii* was enhanced by feeding a diet containing $1 \times 10^7$ CFU/g *E. asburiae* E7 for 28 days, but growth was not improved. Strain E7 can act as an immunostimulant to induce the upregulation of some innate cellular and humoral immune responses, resulting in enhanced resistance to *A. veronii*. Hence, the continuous activation of immune cells can be maintained by adding suitable fresh probiotics to the diet. E7 has the potential to act as a probiotic agent for green, sustainable aquaculture and aquatic product safety.

**KEYWORDS** *Enterobacter asburiae*, common carp, probiotics, antibacterial activity, *Aeromonas*, resistance

**B**acteria are the most common pathogens in farmed fish and cause significant losses in both freshwater and marine aquaculture (1–3). Antimicrobials, especially antibiotics, have been used to control bacterial diseases in fish over the past few decades.

Address correspondence to Fei Ling, feiling@nwsuaf.edu.cn, or Peng-Fei Li, pfli2014@126.com.

The authors declare no conflict of interest.

Although the use of antibiotics is a simple method of controlling aquaculture diseases, it is currently not recommended as a first choice (4, 5) because the application of these chemicals has posed a serious range of hazards on human health, such as the generation of resistant bacteria and the direct or indirect transmission of drug resistance by humans through ingestion in the food chain and exposure to the water environment (6, 7). Hence, it is essential and urgent to find an ecologically efficient and environmentally friendly method of disease control. The use of probiotics as microbial control agents can replace chemical antibiotics and disinfectants for fish disease control (8). Probiotics may promote host health by inhibiting the proliferation of pathogenic microorganisms in the gut, superficial structures, and culture environments, aiding digestion to ensure the optimal utilization of feed, improving the quality of water, and stimulating the host immune system (8, 9). Sequeiros et al. reported that three lactic acid bacteria (LAB) *Lactococcus lactis* TW34, *Carnobacterium* sp. T4, and *L. pentosus* H16 isolated from Patagonian fish effectively decreased the counts of *Vibrio* and *Enterobacter* (10). Zakaria evaluated that *Enterobacter* sp. G87 protected seabass larvae and *Artemia* nauplii from *Vibrio harveyi* infection (11).

Common carp (*Cyprinus carpio*) belongs to the order *Cypriniformes* and the family *Cyprinidae*, which is regarded as the largest family of freshwater fish. Common carp are considered a potential source for commercial aquaculture in many countries because of its high adaptability to the environment and food (12–14). As the third most generally farmed freshwater fish species worldwide, common carp account for more than 80% of fish production in some European countries (15, 16). In 2010, common carp accounted for 9% of total world fish farming production, with Asia accounting for over 90% of the production of common carp. In 2009, China contributed 77% (2,462,346 tons) of total carp farming production (3,216,203 tons) in the world (17). Common carp are usually farmed in a variety of aquaculture systems in Asia, but the most frequent are semi-intensive pond polyculture systems (17). Nevertheless, intensive aquaculture and polluted aquatic environments could result in high mortality due to increased disease incidences, particularly a range of infectious diseases, such as different species of bacterial, viral, and parasitic pathogens (18, 19). Some bacterial diseases cause serious damage to the aquaculture industry. *Aeromonas* species are the main cause of motile *Aeromonas* septicemia (MAS) in fish and are widely found in freshwater environments (20). *Aeromonas* is an opportunistic pathogen that can cause disease in fish, terrestrial animals, and humans (21). These bacteria can induce a variety of human diseases, such as mild diarrhea, life-threatening necrotizing fasciitis, and sepsis (22). In fish, after infection with *Aeromonas*, fish may develop skin discoloration and ulcers, surface bleeding and congestion, abdominal swelling, internal bleeding, and lethality, resulting in high mortality and heavy economic losses to the aquaculture industry (23, 24). To date, there is no appropriate treatment for *Aeromonas* disease in common carp except antibiotics. Therefore, probiotics, as one of the alternative strategies to antibiotics, are urgently needed and necessary for the common carp industry to be ecologically efficient and environmentally friendly.

Probiotics are defined by the World Health Organization (WHO) as live microorganisms that can enhance vitality, strengthen immunity, promote digestive system function, and confer health benefits to the host when consumed in suitable doses (25). A crucial criterion for screening probiotics is that they should not be harmful to host health. Consequently, hosts should be challenged by candidate strains under stress-free situations to assay probiotic pathogenicity (26). Another significant criterion for screening a probiotic is assessing its sensitivity to antibiotics used by humans and animals because the emergence of resistant bacteria is an important food safety hazard (27). In addition to nonpathogenic properties, ideal probiotics need to be tolerant to acids and bile salts to arrive in the gut completely and remain active (28). Furthermore, high temperature tolerance allows probiotics to withstand extreme stress and facilitates industrial-scale production and formulation preservation (29). The origin of probiotics is a fundamental element in the selection of probiotics, which directly influences the effectiveness of probiotics in actual production. The intestine is an essential

**TABLE 1** The diameter of inhibition zone of *E. asburiae* E7 for common aquatic pathogens

| Indicator bacteria | Bacteriostatic circle (mm)[a] |
| --- | --- |
| *Aeromonas hydrophila* | 8.76 |
| *Aeromonas veronii* | 8.34 |
| *Aeromonas caviae* | 8.45 |
| *Aeromonas media* | 8.11 |
| *Aeromonas jandaei* | 9.24 |
| *Aeromonas enteropelogenes* | 8.32 |
| *Aeromonas schubertii* | 8.51 |
| *Aeromonas salmonicida* | 8.47 |
| *Pseudomonas aeruginosa* | 12.05 |
| *Pseudomonas putida* | 8.80 |
| *Plesiomonas shigelloides* | 9.67 |
| *Shewanella* sp. | 8.50 |
| *Vibrio alginolyticus* | No bacteriostatic circle |
| *Vibrio cholerae* | No bacteriostatic circle |
| *Acinetobacter johnsonii* | No bacteriostatic circle |
| *Citrobacter braakii* | No bacteriostatic circle |

[a]The total size of the inhibition zone is shown; the original diameter of the paper discs was 6 mm.

physiological barrier and the most frequent source of probiotics in aquaculture (30). Some studies have demonstrated that probiotics isolated from the gut of fish have the capacity to suppress the proliferation of pathogens and are more likely to predominate in the intestinal environment (31, 32). As such, the gastrointestinal (GI) tract and the mucous of fish are the most common sources of probiotics in aquaculture (26, 33). Host-derived probiotics are far superior to other environmentally selected probiotics when it comes to safety and efficacy (34). A wide range of microorganisms have been described as probiotics used in aquaculture. Gram-positive LAB or *Bacillus* spp. are the most commonly used candidate probiotics (35). There are few reports on Gram-negative probiotics, so screening of new Gram-negative species to enrich probiotic libraries is needed.

In the current research, a promising probiotic strain named E7 was isolated from healthy common carp intestines and identified as *Enterobacter asburiae* by morphology, physiology and biochemistry, molecular biology, and bioinformatics. *In vitro* antimicrobial activity, biosafety, and physical tolerability were evaluated. In addition, the aim of this study was to evaluate the potential of *E. asburiae* E7 as a feed probiotic for growth, immunity, and resistance to *A. veronii* infection in common carp.

## RESULTS

**Probiotics isolation, characterization, and identification.** A total of 151 strains of bacteria were isolated and purified from the intestines of 10 healthy common carp for the following *in vitro* antagonism assay. Twenty-seven strains were antagonistic to common aquatic pathogens (Table 1), with one strain named E7 having broad-spectrum antimicrobial activity (Fig. 1). The strain E7 grew well on Luria-Bertani (LB), *Enterococcus* agar (EA), Pfizer *Enterococcus*-selective agar (PSE), freshwater fish agar (FWA), *Bacillus megatherium* medium (BM), tryptone soy agar (TSA), brain heart infusion (BHI) and MRS medium, but a small numbers in MRS. The growth curve of E7 in LB broth medium is shown in Fig. S1 in the supplemental material.

The colonies formed after 24 h of incubation at 28°C were 2 to 4 mm in diameter and had typical colony characteristics, including pale yellow or milky white in color and round with a smooth surface and regular edges (Fig. 2A). This bacterial strain was Gram negative, rod shaped or short rod shaped, and spore free (Fig. 2B). According to the physiological and biochemical tests (Table 2), strain E7 was motile and could produce amino acid decarboxylase to decompose amino acids into amine (lysine to cadaverine and ornithine to putrescine). E7 could ferment a variety of sugars and alcohols (melibiose, raffinose, mannitol, and sorbitol) and could also use citrate. Both the methyl red test and the Voges-Proskaurer (V-P) test were negative, suggesting that the strain is unable to produce pyruvate through the catabolism of glucose.

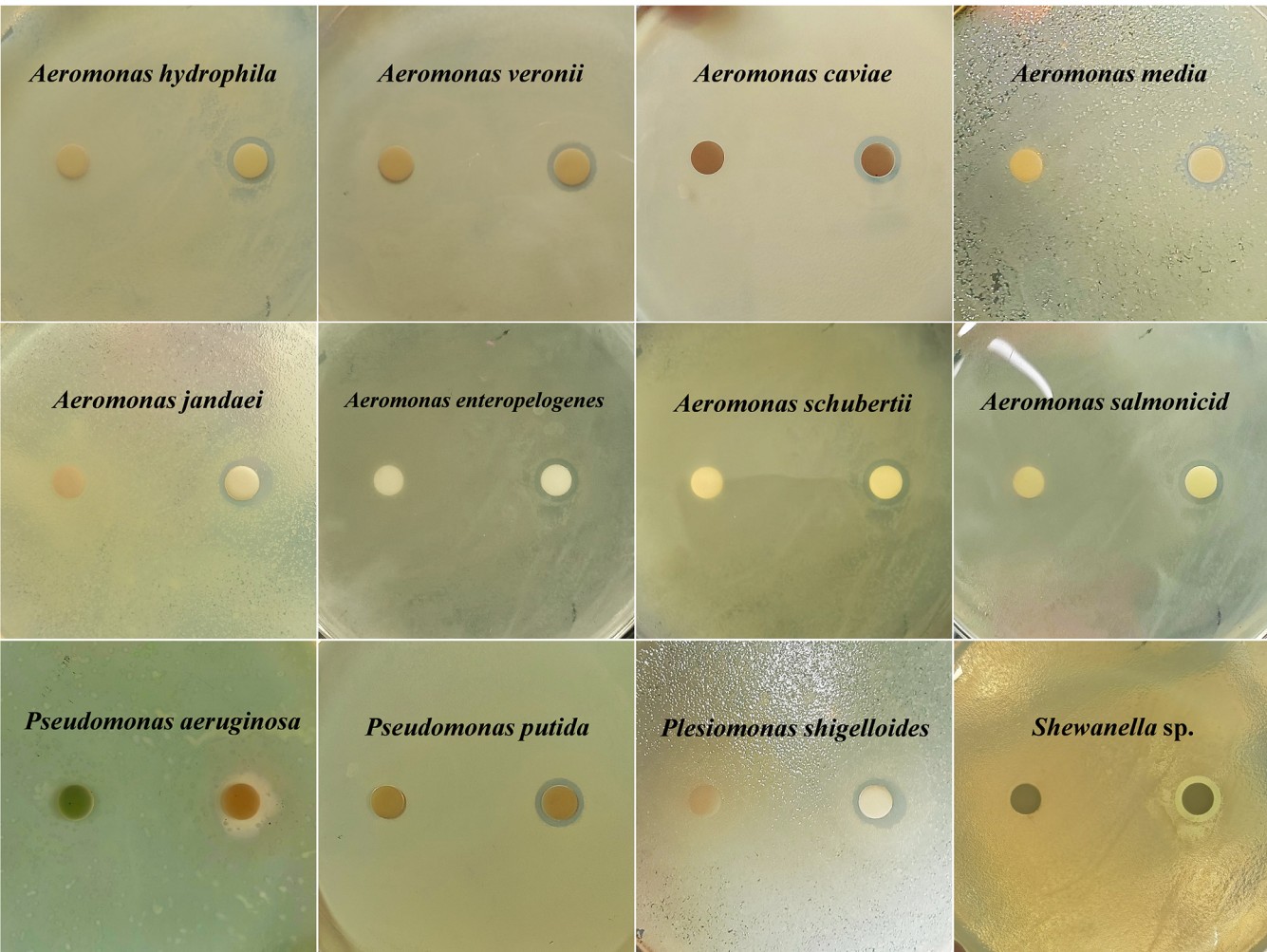

**FIG 1** Antibacterial activity of *E. asburiae* E7 against common aquatic pathogenic bacteria *Aeromonas hydrophila*, *A. veronii*, *A. caviae*, *A. media*, *A. jandaei*, *A. enteropelogenes*, *A. schubertii*, *A. salmonicida*, *Pseudomonas aeruginosa*, *Ps. putida*, *Plesiomonas shigelloides*, and *Shewanella* sp. Disks (6 mm in diameter) were impregnated with 10 µL of sterile PBS (left) and 10 µL of the E7 suspension ($1 \times 10^7$ CFU/mL; right), respectively. Pathogenic bacteria as indicators were coated on LB agar plates.

The partial 16S rRNA sequence comparison of E7 displayed 96 to 97% sequence similarity to the *Enterobacter* genus. A maximum likelihood (ML) phylogenetic tree based on 16S rRNA gene sequences revealed that strain E7 did not cluster into any branch (Fig. 3A). Thus, the above data indicated that the 16S rRNA gene was unable to classify E7 to the species level and was only identified as *Enterobacter* sp. This strain was further characterized by the *gyrB* gene sequence, which displayed 97.64% similarity to *E. asburiae* En30 (AP024498.1). In addition, an ML phylogenetic tree of the *gyrB* gene sequences revealed that strain E7 closely clustered with *E. asburiae* (Fig. 3B). The 16S rRNA and *gyrB* sequences of E7 were submitted to GenBank with the accession numbers OP522325 and OP558470, respectively. On the basis of morphological, biochemical, and molecular characteristics, isolate E7 was identified as *E. asburiae* E7.

**Antimicrobial activity.** The agar diffusion test was used for determining the antibacterial activity of *E. asburiae* E7. There was excellent antibacterial activity of strain E7 against Gram-negative pathogens, including *Aeromonas hydrophila*, *A. veronii*, *A. caviae*, *A. media*, *A. jandaei*, *A. enteropelogenes*, *A. schubertii*, *A. salmonicida*, *Pseudomonas aeruginosa*, *Ps. putida*, *Plesiomonas shigelloides*, and *Shewanella* (Fig. 1); there was an absence of antagonistic activity of E7 against pathogens *Vibrio alginolyticus*, *V. cholerae*, *Acinetobacter johnsonii*, *Citrobacter braakii* (Table 1).

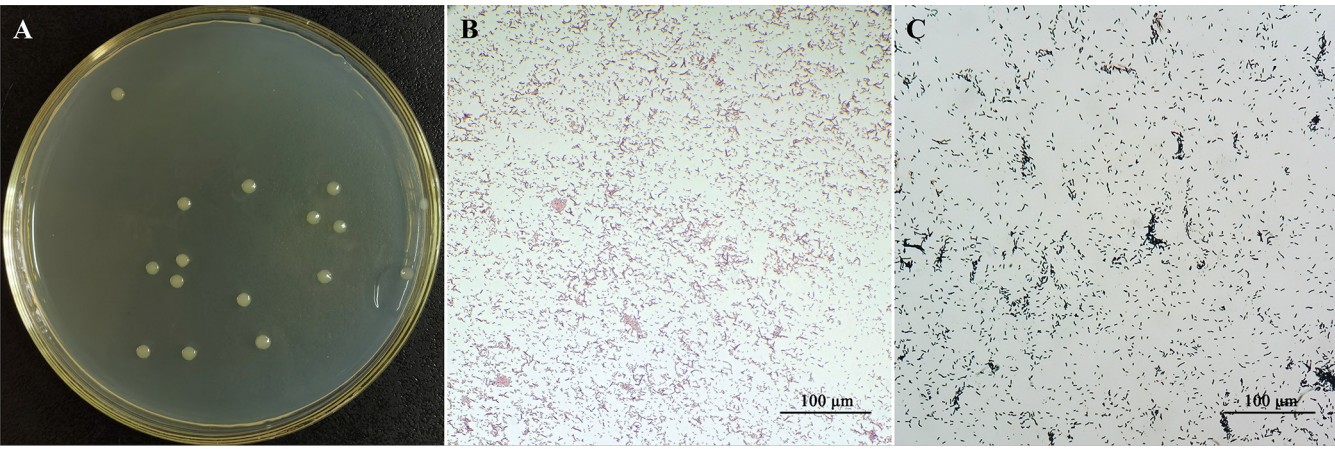

**FIG 2** Morphological characteristics of *E. asburiae* E7. (A) Colony morphology of E7 on LB agar plates after incubation at 28°C for 24 h. (B) Gram stain. (C) Gram-positive control. Gram staining was performed after bacteria had been cultured for 6 h (logarithmic phase of growth), and photographs were taken with a ×40 lens objective.

**Safety evaluation.** *E. asburiae* E7 exhibited no hemolytic activity after 24 h of incubation at 28°C on goat blood agar plates, whereas the positive control, *A. veronii*, had a clear hemolytic loop (Fig. 4A). Additionally, in fish injected intraperitoneally (i.p.) with 0.1 mL of *E. asburiae* E7 suspension ($1 \times 10^9$ CFU/mL), no significant differences in survival within 7 days compared to the control group (Fig. 4B).

**Histopathological examination.** Histopathological analysis was performed on four fish organs. Compared with the control group, no pathological abnormality was observed in treated fish (Fig. 4C). Sections of liver tissue indicated normal structure and arrangement of hepatic parenchyma. The intact nucleus of hepatocytes is located in the middle. Spleen sections revealed normal characteristics, with intact splenic cord and sinusoid. Kidney sections displayed clear and complete structures of glomeruli and renal tubules. Intestinal villi cells showed intact features and were arranged in a classical finger-shaped appearance.

**Antibiotic sensitivity.** The results of the antibiotic sensitivity test showed that strain E7 was susceptible to spectinomycin, tobramycin, kanamycin, gentamicin, streptomycin, chloramphenicol, cefepime, ceftriaxone, cefoperazone, ofloxacin, norfloxacin, ciprofloxacin, levofloxacin, tetracycline, amikacin, piperacillin, nitrofurantoin, aztreonam,

**TABLE 2** Physiological-biochemical characteristics of the antagonistic *E. asburiae* E7

| Characteristics | E7 |
| --- | --- |
| Shape | Rod shape |
| Gram stain | − |
| Motility | + |
| Oxidase | − |
| Lysine | + |
| Ornithine | + |
| Citrate | + |
| H$_2$S | − |
| Urease | − |
| Indole | − |
| Methyl red | − |
| Phenylalanine | − |
| V-P | − |
| Mannitol | + |
| Inositol | − |
| Sorbitol | + |
| Melibiose | + |
| Adonitol | − |
| Raffinose | + |

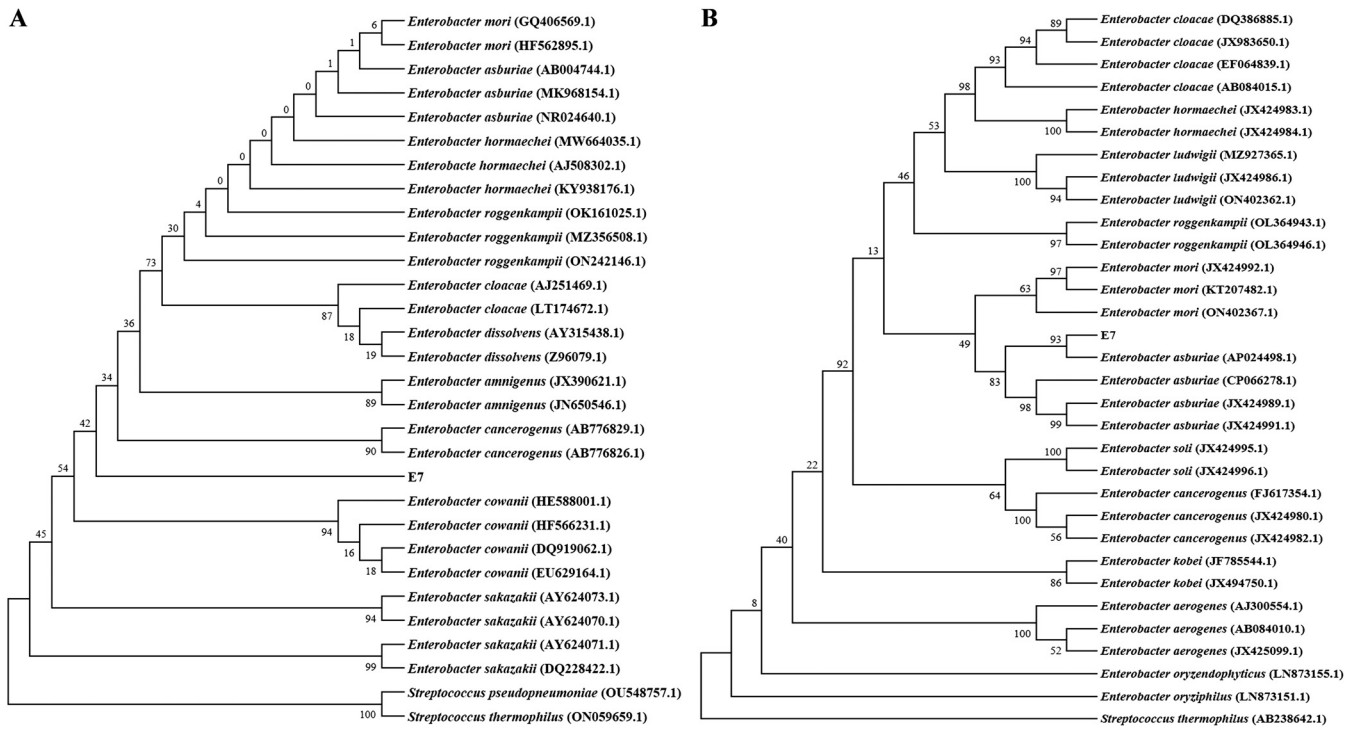

**FIG 3** Maximum likelihood phylogenetic tree of *E. asburiae* E7. (A and B) Phylogenetic trees of E7 based on 16S rRNA gene sequences (A) and *gyrB* gene sequences (B) are shown. The gene sequences from 30 related strains were aligned using the ClustalW tool in MEGA 7.0.26. Phylogenetic analysis was performed using the maximum likelihood method with 1,000 replications in bootstrap test; the level of bootstrap support is shown at all nodes.

Bactrim, enrofloxacin, sulfamethoxazole, sulfachlorpyridazine, oxytetracycline, doxycycline hydrochloride, chlortetracycline hydrochloride, thiamphenicol, florfenicol, apramycin sulfate, and neomycin sulphate and was resistant to vancomycin, midecamycin, clindamycin, clarithromycin, penicillin, erythromycin, cefoxitin, cefazolin, cefalotin, cefotaxime, cefuroxime, minocycline, ampicillin, and oxacillin (Table 3).

**Antibiotic resistance genes (ARGs).** PCR coupled with agarose gel electrophoresis showed that E7 was missing most of the ARGs, but *sul2*, *tetA*, and *blaCMY* were present (Fig. S3).

**pH, temperature, and bile salt tolerance.** *E. asburiae* E7 was found to be viable at pH 4 and grew well quantitatively and morphologically under experimental culture conditions at pH values of 5 to 7 (Fig. 5A and D). In the temperature tolerance test, E7 could grow from 28 to 40°C and partially survived to 45°C (Fig. 5B and E). Furthermore, the results showed that E7 was highly tolerant to bile salts and even grew at high bile salt concentrations of 4%; in addition, the number of bacteria did not differ significantly in the treatments with 2 to 4% bile salt (Fig. 5C and F).

**Influence of *E. asburiae* E7 on the growth of common carp.** The effects of *E. asburiae* E7 diet administration on the growth performance of common carp are shown in Fig. 6. During the 28-day feeding period, the body length, body weight, and carcass weight of the treatment groups (*E. asburiae*) decreased slightly compared with the controls, but the difference was not statistically significant.

**Serum biochemistry index.** After 7, 14, 21, and 28 days of feeding, there was no significant difference in acid phosphatase (ACP) (Fig. 7A) and total antioxidant capacity (T-AOC) (Fig. 7D) activities between the treatment and control groups, respectively. Nevertheless, there was significantly higher alkaline phosphatase (AKP) activity ($P < 0.05$) (Fig. 7B) and total protein (TP) levels ($P < 0.05$) (Fig. 7E) than the controls at the third week of feeding. Interestingly, glutathione (GSH) activity was significantly downregulated within 21 days of feeding E7 compared with the controls ($P < 0.01$), while

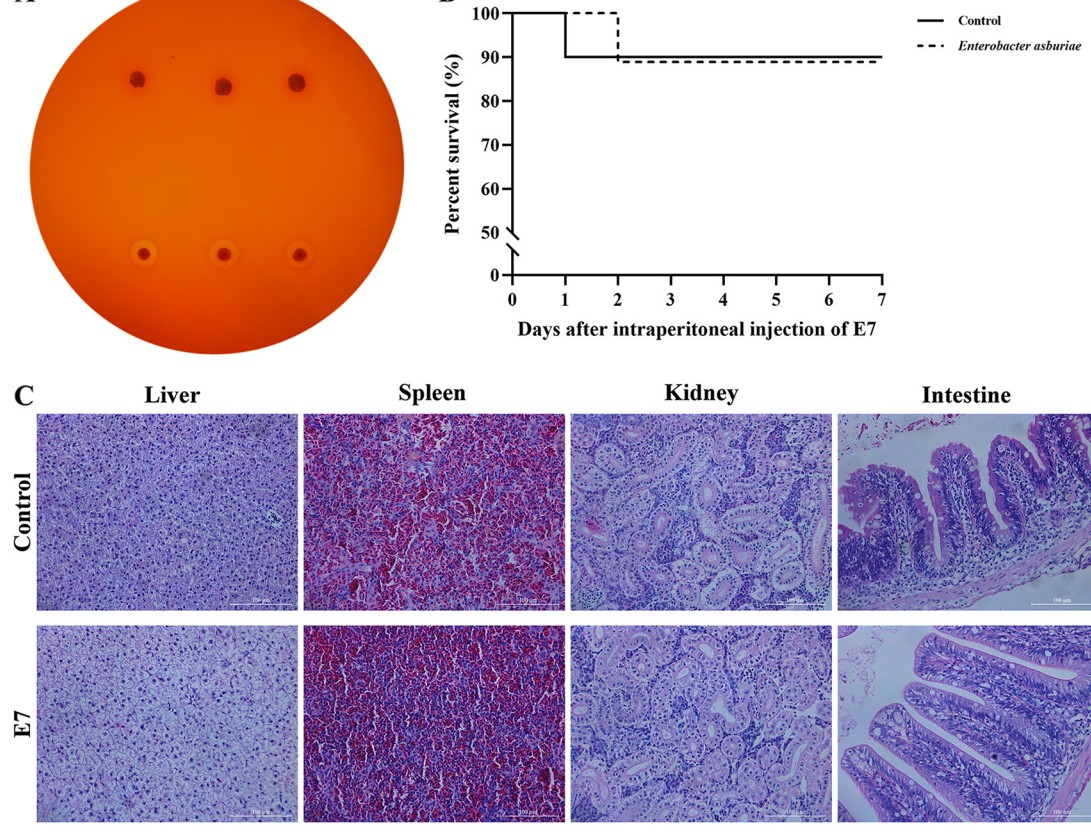

**FIG 4** Safety evaluation of *E. asburiae* E7. (A) Hemolysis tests of *E. asburiae* E7 (top) and positive-control *A. veronii* (bottom) were performed on goat blood plates. A positive reaction is indicated by a clear zone around the colony, and a negative reaction was defined as the absence of a hemolytic zone around the colony. (B) Percent survival of common carp injected intraperitoneally with *E. asburiae* E7. The fish (*n* = 10) were injected intraperitoneally with 0.1 mL of strain E7 suspension at a concentration of $1 \times 10^9$ CFU/mL or sterile PBS (control) for 7 days, and percent survival was calculated from the daily number of deaths. There was no significant difference between the treatment group (*E. asburiae*) and the control group (*P* > 0.05). (C) Histopathological examination of the liver, spleen, kidney, and intestine was performed on common carp after 28 days of E7 feeding. The control group was fed regular diets; scale bar, 100 $\mu$m.

GSH activity was significantly upregulated at 28 days (*P* < 0.05) (Fig. 7C). Analogously, serum superoxide dismutase (SOD) activity was significantly lower than that observed in the control groups in the second and fourth weeks of feeding but was transiently significantly upregulated during the third week (*P* < 0.01) (Fig. 7F).

**Gene expression profile.** The relative mRNA expression of immune-related genes in the kidneys of common carp fed with a diet containing strain E7 is displayed in Fig. 8, including interleukin-1$\beta$ (*IL-1$\beta$*), interleukin-10 (*IL-10*), interleukin-8 (*IL-8*), interferon (*IFN*), tumor necrosis factor-$\alpha$ (*TNF-$\alpha$*), lysozyme, and transforming growth factor-$\beta$ (*TGF-$\beta$*).

According to the results, *IL-1$\beta$*, *IFN*, and *TNF-$\alpha$* were observed to be significantly upregulated after week 4 of feeding (*P* < 0.01) (Fig. 8A, D, and E). Likewise, the mRNA expression of *IL-10*, *IL-8*, and *lysozyme* were persistently upregulated throughout the 28-day feeding period, and significant differences were observed at the first, third, and fourth weeks (*P* < 0.01) (Fig. 8B, C, and F). In addition, there was a significant increase in the mRNA expression of *TGF-$\beta$* (*P* < 0.01) (Fig. 8G) between fish treated with the *E. asburiae* E7 diet and controls at the third week of feeding.

**Disease resistance against *Aeromonas veronii*.** Mortality of common carp was documented daily for 8 days after challenge with *A. veronii* to assess the protective efficacy of *E. asburiae* E7 against bacterial infection in fish, and the survival curves are presented in Fig. 9. The percent survival rate of fish in the control group was 54% compared with 91.05% in the group treated with $10^7$ CFU/g E7. Statistical analysis demonstrated

**TABLE 3** Results of antibiotic sensitivity testing for *E. asburiae* E7

| Antibiotic | Sensitivity[a] | Antibiotic | Sensitivity[a] |
|---|---|---|---|
| Vancomycin | R | Ciprofloxacin | S |
| Midecamycin | R | Levofloxacin | S |
| Spectinomycin | S | Tetracycline | S |
| Clindamycin | R | Amikacin | S |
| Clarithromycin | R | Minocycline | R |
| Tobramycin | S | Ampicillin | R |
| Kanamycin | S | Piperacillin | S |
| Gentamicin | S | Oxacillin | R |
| Penicillin | R | Nitrofurantoin | S |
| Erythromycin | R | Aztreonam | S |
| Streptomycin | S | Polymyxin | IR |
| Chloramphenicol | S | Bactrim | S |
| Cefoxitin | R | Enrofloxacin | S |
| Cefazolin | R | Sulfamethoxazole | S |
| Cefalotin | R | Sulfachlorpyridazine | S |
| Cefepime | S | Oxytetracycline | S |
| Cefotaxime | R | Doxycycline hydrochloride | S |
| Ceftriaxone | S | Chlortetracycline hydrochloride | S |
| Cefuroxime | R | Thiamphenicol | S |
| Ceftazidime | IR | Florfenicol | S |
| Cefoperazone | S | Apramycin sulfate | S |
| Ofloxacin | S | Neomycin sulphate | S |
| Norfloxacin | S | Bacitracin zinc | IR |

[a]According to the inhibition zone diameters, antibiotic sensitivity was expressed in terms of susceptible (S), resistant (R), or intermediate resistant (IR).

that dietary supplementation with strain E7 significantly enhanced the resistance of common carp to *A. veronii* infection ($P < 0.01$) (Fig. 9).

## DISCUSSION

Microbial interventions play a crucial role for aquaculture, and efficient probiotic treatments can provide wide-spectrum and superior nonspecific protection against disease (36). The first trial to apply probiotics to aquaculture used a commercial feed designed for terrestrial animals in 1986. Spores of *Bacillus cereus* var. toyoi isolated from soil decreased the mortality of Japanese eel infected with *Edwardsiella* sp. Likewise, supplementation in the feed improved the growth rate of yellowtail (37). The use of aquatic probiotics is a relatively new concept. Currently, various bacterial taxa are already being accessed as probiotics for aquatic animals, but the search for new microorganisms that can be used as aquatic probiotics is ongoing.

When studying probiotics for aquatic use, it is vital to account for a number of influences that differ from terrestrial probiotics. Terrestrial and aquatic animals differ greatly in the level of interaction between their gut microbiota and surrounding environment. Gram-positive obligate or facultative anaerobes dominate the GI microbiota of humans and land-farmed animals. The majority of probiotics belong to dominant or subdominant genera within these microflorae, such as *Lactobacillus*, *Bifidobacterium*, and *Streptococcus* (38). In fish and shellfish, Gram-negative facultative anaerobes predominate in the digestive tract (38). *Vibrio* and *Pseudomonas* are the most common genera in crustaceans (39), marine fish (40), and bivalves (41). *Aeromonas*, *Plesiomonas*, and *Enterobacteriaceae* are dominant in freshwater fish (40). Accordingly, the most effective probiotics in aquaculture may differ from terrestrial species due to the specificity of aquatic microbiota. For instance, bacteriocins produced by LAB are normally only effective against closely related species, and most pathogens in aquaculture are Gram-negative bacteria, so bacteriocins from LAB may not inhibit fish pathogens (42). Hence, it is imperative to develop special probiotics for aquatic products.

*E. asburiae* was first described in 1986 by Brenner et al. (43) from strains of the enteric group 17. Sutton et al. provided evidence that *E. muelleri* is a later heterotypic

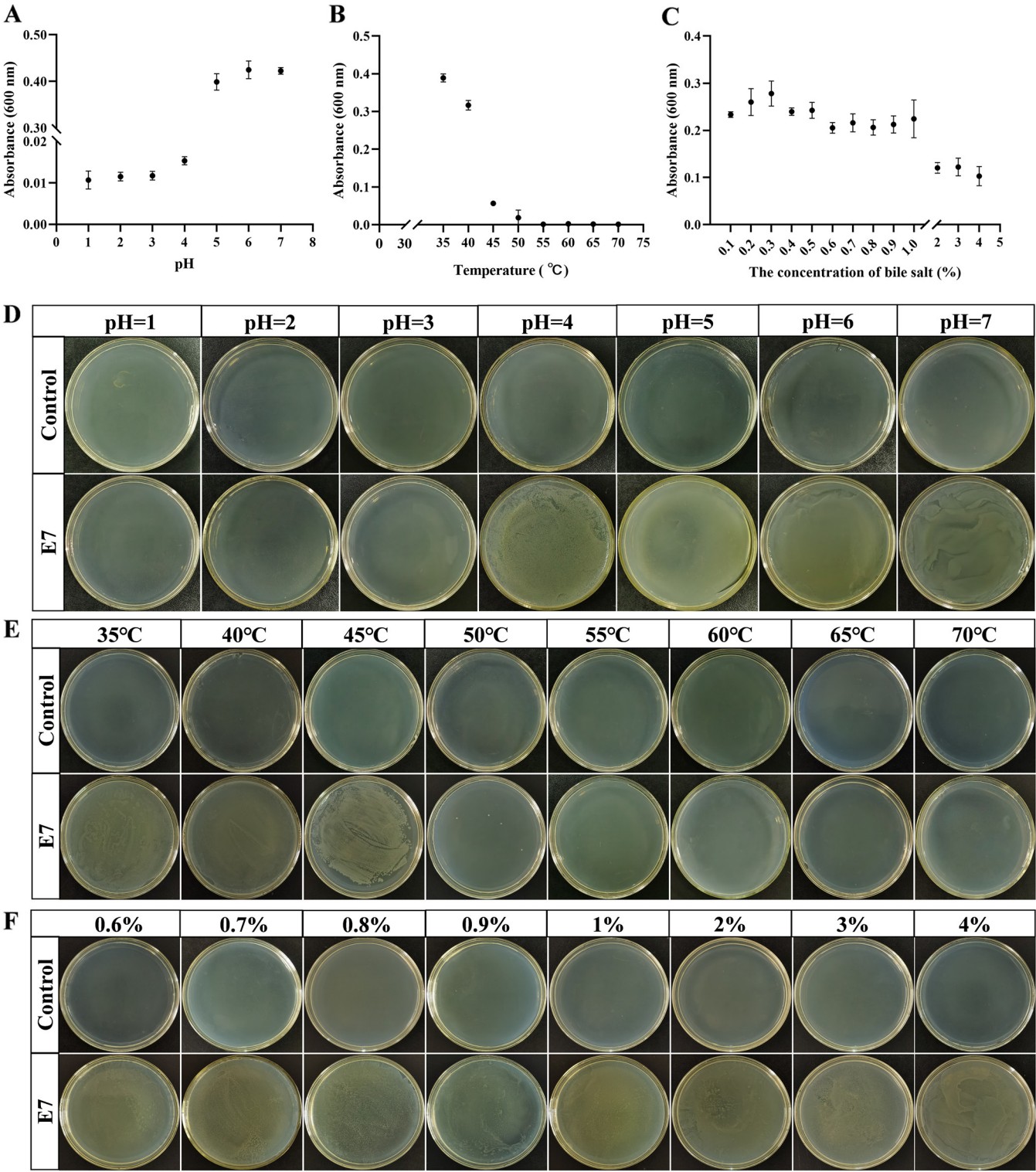

**FIG 5** pH, temperature, and bile salt tolerance of *E. asburiae* E7. (A and D) The effect of pH on the growth of *E. asburiae* E7. (B and E) Temperature tolerance of E7. (C and F) Bile salt tolerance of E7. Activated *E. asburiae* E7 ($1 \times 10^7$ CFU/mL) was inoculated at 1% (vol/vol) in LB broth medium with different pH values (1, 2, 3, 4, 5, 6, and 7), temperatures (35, 40, 45, 50, 55, 60, 65, and 70°C), and bile salt concentrations (0.1, 0.2, 0.3, 0.4, 0.5, 0.6, 0.7, 0.8, 0.9, 1, 2, 3, and 4%, wt/vol) and incubated for 24 h. The absorbance at a wavelength of 600 nm was measured using 0.2 mL of the bacterial solution. At the same time, 0.1 mL of bacterial culture was spread on LB agar plates and incubated at 28°C for 24 h, and photographs were taken. Data represent mean ± standard deviation (SD; error bars) of 27 replicates.

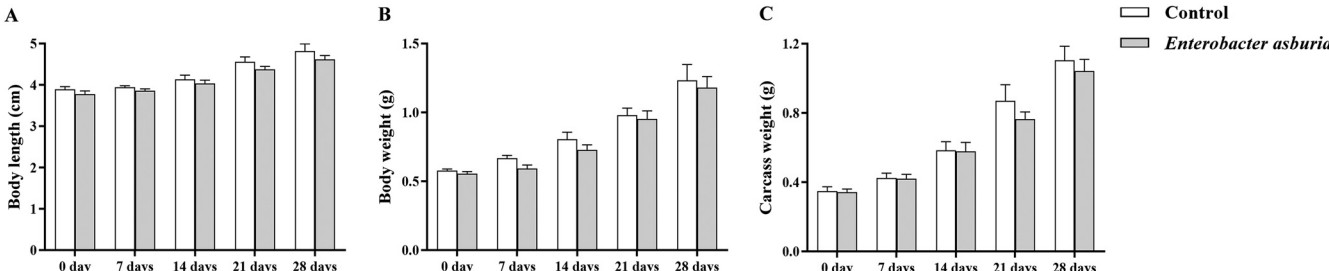

**FIG 6** Growth performance of dietary supplementation of *E. asburiae* E7 in common carp. (A to C) Body length (A), body weight (B), and carcass weight (C) were measured every 7 days after fish were fed daily with a regular diet containing $1 \times 10^7$ CFU/g *E. asburiae* E7 at 4% of body weight for 28 days; the control group was fed a regular diet. Data represent mean ± SD ($n = 45/n = 18$). Forty-five fish in each group were randomly selected to measure body length and body weight, and 18 of them were used to measure carcass weight. Statistical analyses were performed using independent-samples *t* tests. There was no significant difference between the treatment group (*E. asburiae*) and the control group ($P > 0.05$).

synonym of *E. asburiae* based on computational analysis of sequenced *Enterobacter* genomes (44). Recently, *E. asburiae* has been widely reported in industrial production, agricultural field and environmental remediation, such as hydrogen production (45), wastewater treatment, algae removal (46), carcinogenic dye degradation (47), induction of salt tolerance traits in plant rhizospheres, nitrogen fixation, and promotion of plant rhizosphere growth (48). However, there are also some strains isolated from the clinic that show pathogenicity as opportunistic pathogens (49, 50). There are few reports on *E. asburiae* as probiotics in aquaculture. In a recent study, Tang et al. isolated *E. asburiae* C28 from the intestine of *Carassius auratus*, which reduced the load of potential pathogens, increased the amount of potential probiotics in the host gut, and decreased the death rate of *C. auratus* challenged by *A. hydrophila* (51). He et al. reported an isolate *E. asburiae* X8 from the gut of grass carp with the ability to produce cellulase (52).

Comparative analysis of the 16S rRNA sequence is the most common genotype identification method for bacteria (53). Strains displaying greater than 97% sequence similarity in 16S rRNA are usually regarded as the same species (54). However, due to

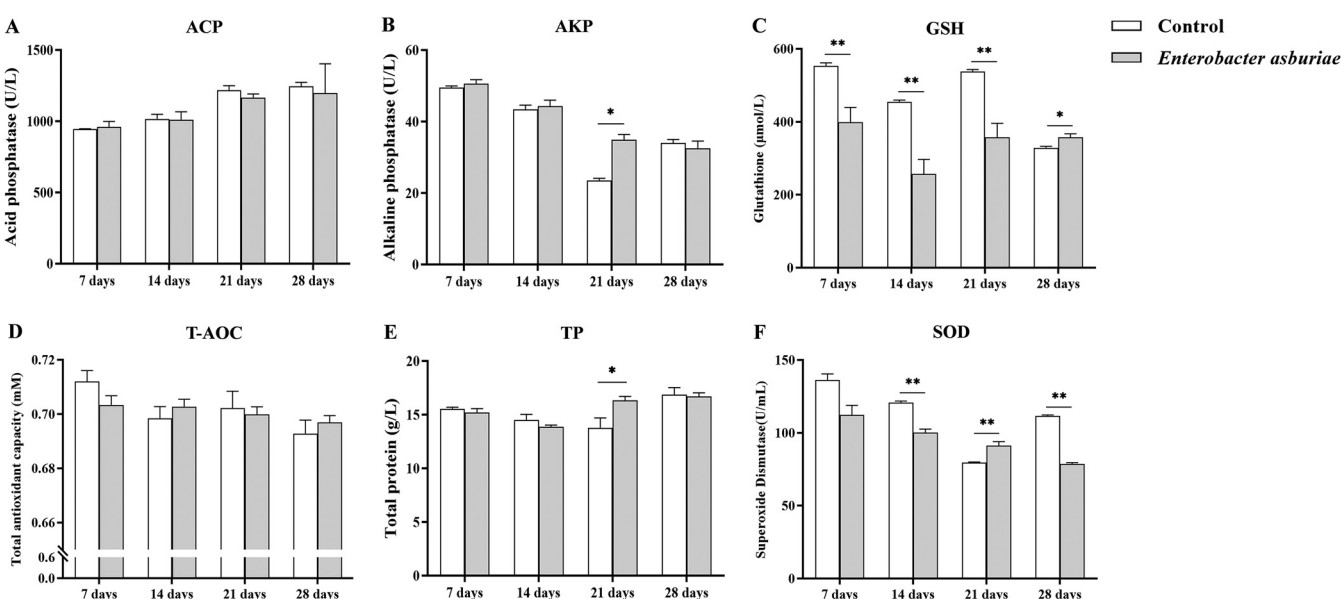

**FIG 7** Serological immune index of the common carp with *E. asburiae* E7 supplementation. (A to F) Acid phosphatase activity (A), alkaline phosphatase activity (B), glutathione level (C), total antioxidant capacity (D), total protein level (E), and superoxide dismutase activity (F) were detected every 7 days after fish were fed daily with a regular diet containing $1 \times 10^7$ CFU/g *E. asburiae* E7 at 4% of body weight for 28 days; the control group was fed a regular diet; ACP, acid phosphatase; AKP, alkaline phosphatase; GSH, glutathione; T-AOC, total antioxidant capacity; TP, total protein; SOD, superoxide dismutase. Values represent mean ± SD ($n = 18$). Statistical analyses were performed using independent-samples *t* tests. The single asterisks (*, $P < 0.05$) and double asterisks (**, $P < 0.01$) indicate statistical significance compared with controls.

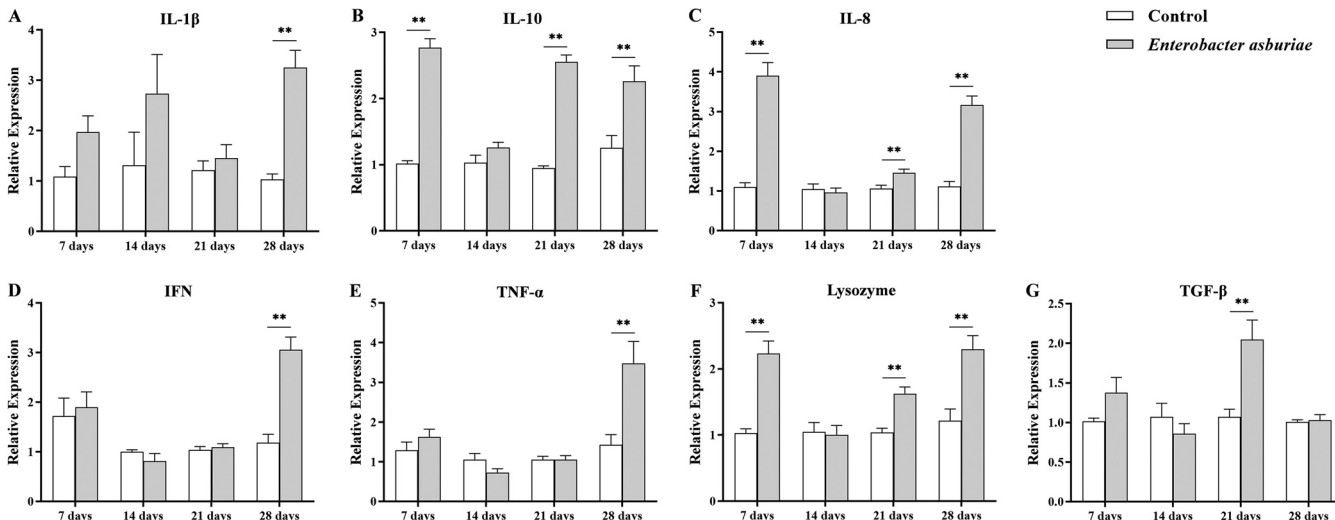

**FIG 8** Expression of immune-related genes in the kidneys of common carp. (A to G) The relative expression levels of *IL-1β* (A), *IL-10* (B), *IL-8* (C), *IFN* (D), *TNF-α* (E), *Lysozyme* (F), and *TGF-β* (G) were detected by RT-qPCR, and values were normalized against *β-actin* as a reference gene. Kidneys of fish were collected every 7 days after carp were fed daily with a regular diet containing $1 \times 10^7$ CFU/g *E. asburiae* E7 at 4% of body weight for 28 days; the control group was fed a regular diet. Values are expressed as mean ± SD ($n = 18$). Statistical analyses were performed using independent-samples *t* tests. The single asterisks (*, $P < 0.05$) and double asterisks (**, $P < 0.01$) indicate statistical significance compared with control.

the high degree of conservation of this gene, phylogenetic analysis based on the 16S rRNA gene alone is not sufficient to distinguish *Enterobacter* species. The present results are the same as above. An ML phylogenetic tree based on 16S rRNA gene sequences indicated that E7 was not clustered into any branch (Fig. 3A). In comparison, the *gyrB* gene, encoding the subunit B protein of DNA gyrase (a type II DNA topoisomerase), has a critical role in DNA replication and is widely distributed in bacterial species (55). Generally, the sequence of *gyrB* is considered to be more discriminating than the 16S rRNA gene (56), so the *gyrB* gene had been proposed to be an outstanding molecular marker candidate for identifying bacterial species (57). In the current research, a phylogenetic tree based on *gyrB* gene sequences revealed that E7 clustered with the *E. asburiae* clade (Fig. 3B). Culture characteristics and physiological and biochemical tests of bacteria can initially identify bacteria. The isolate E7 was identified with reference to standard strain results and Bergey's Manual of Determinative Bacteriology (58). The strain in this study exhibits similar physiological and biochemical characteristics as *E. asburiae* reported by Anne et al. (59). Coupled with multigene sequence (16S rRNA and *gyrB* gene) analyses, morphological, physiological, and biochemical characterization identified the isolate E7 as *E. asburiae*.

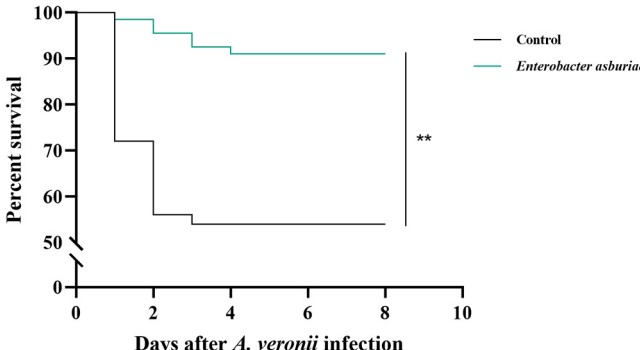

**FIG 9** Effect of *E. asburiae* E7 supplementation on survival of common carp challenged with *A. veronii*. Fish were fed with regular diets and $10^7$ CFU/g *E. asburiae* E7-containing diets at 4% of body weight for 28 days, followed by an intraperitoneal injection of 0.1 mL of *A. veronii* at a concentration of $1 \times 10^7$ CFU/mL. Survival was monitored for 8 days ($n = 50$). Asterisks indicate statistical significance compared with control ($P < 0.01$).

Selection standard for probiotics is supposed to assess the ability of bacterial strains to restrict or disturb pathogen development. The majority of studies selecting microorganisms as probiotic candidates concentrated on *in vitro* antagonism assays with pathogens that were exposed to candidate probiotics or their extracellular products (60). Current *in vitro* studies demonstrated that *E. asburiae* E7 displayed antagonistic activity against pathogens of the genus *Aeromonas*, including mesophilic motile *Aeromonas* (*Aeromonas hydrophila*, *A. veronii*, *A. caviae*, *A. media*, *A. jandaei*, *A. enteropelogenes*, and *A. schubertii*) and psychrophilic nonmotile *Aeromonas* (*A. salmonicida*). In addition, E7 could also antagonize other common fish pathogens, such as *Pseudomonas aeruginosa*, *Ps. putida*, *Plesiomonas shigelloides*, and *Shewanella*. So far, no similar results have been reported for *E. asburiae*. According to some researchers, the antibacterial property of probiotics may be related to nutrition competition or inhibitory substances produced by bacteria, such as broad-spectrum antibiotics, bacteriocins, organic acids, siderophores, hydrogen peroxide, and enzymes (61). However, the mode of action by which the antagonism of E7 occurs in this study is unclear. Another most important criterion for the selection of probiotics is that the probiotics should not cause host disease, which can be confirmed by *in vivo* testing. The isolate E7 used in this study was not pathogenic in common carp, as no mortality or morbidity was observed after intraperitoneal injection and no hemolytic activity was observed in blood plates. More importantly, no pathological changes were observed in the main organs (liver, spleen, kidney, and intestine) of fish after 28 days of E7 feeding. In addition, the biological and ecological security of potential probiotic strains is crucial in the selection process. Consequently, strains should be tested for resistance to various common antibiotic classes (e.g., tetracycline, macrolides, and quinolones) and subsequent confirmation that resistance genes or virulence plasmids have not been transmitted (38). The results of antibiotic susceptibility testing of strain E7 showed that this strain was sensitive to the majority of antibiotics used in clinical treatment in humans and to commonly used aquatic/veterinary antibiotics. In general, E7 was sensitive to tetracycline, aminoglycoside, phenicol, 4-quinolone, and sulfonamide antibiotics and was resistant to partial macrolide and $\beta$-lactam antibiotics. Meanwhile, E7 lacked aminoglycoside, chloramphenicol, macrolide, and quinolone ARGs, which was consistent with antibiotic susceptibility testing. However, E7 carried part of the sulfonamide resistance gene *sul2*, tetracycline resistance gene *tetA*, and $\beta$-lactam resistance gene *blaCMY*. This phenomenon could explain why E7 was resistant to individual antibiotics of a certain class. To summarize, E7 has the potential to be exploited as a probiotic preparation due to its ability to antagonize pathogens, the fact that it is nonpathogenic to the host, and its ecological security.

The origin of probiotics is a crucial element affecting their function and efficacy. Microorganisms exhibit different physiological or biochemical activities during development depending on the environment (freshwater or seawater) and source. In terms of security and effectiveness, probiotics from hosts are far better than those screened from other environments (34). To ensure the expected effect, probiotics isolated from the host ought to not only suppress pathogens but also adapt to the gut environment to maximize their beneficial effects for the host (62). Heavy acidic conditions in the GI tract may have a negative impact on bacteria. For this reason, it is only when probiotics can survive under low-pH conditions in the fish gut that they can enhance the immune response of the host (63). The results of the tolerance capacity assay in this study showed that strain E7 was resistant to acidic conditions and high concentrations of bile salts. Analogous results were also reported by Koyama et al. (64), who demonstrated that *E. faecium* IS-27526 was tolerant to acid and bile. Furthermore, strain E7 was able to survive at 45°C and grew well on LB, EA, PSE, FWA, BM, TSA, and BHI medium. These characteristics indicate that E7 has strong environmental adaptability and can take in different nutritional sources, which is conducive for large-scale production and application. Therefore, E7 has the value of being developed as an aquatic probiotic.

Research on diets containing probiotics indicated that another important role of probiotics is to improve feed efficiency and promote growth. Varela et al. reported that the addition of probiotic strain Pdp11 to the diet of *Sparus auratus* helped to promote

growth and improve stress tolerance to high stocking density (65). Probiotic diets containing *Vibrio* sp. CC8 and *B. cereus* CC27, alone or in combination, could improve growth performance in juvenile Nile tilapia *Oreochromis niloticus* (66). In this study, dietary supplementation with E7 had no significant effect on growth and survival of common carp, indicating that it may exert probiotic effects on the host in other ways.

Several investigators reported that probiotics could stimulate and modulate the innate immune system of aquatic animals (67). Blood parameters reflect the health status, nutrient metabolism, and immune response of fish, which help to determine the influence of dietary supplements on the health status of fish (68). As a core compound of lysosomal enzymes, ACP plays a critical part in the immune system and is recognized as a marker of macrophage activation in creature models (69). AKP is reported as an extracellular enzyme that hydrolyzes phosphate binders in a variety of organic compounds, including proteins, lipids, and carbohydrates (70). Sea cucumbers supplemented with *L. plantarum* LL11 diets revealed higher levels of ACP and AKP, as reported by Li et al. (71). Plasma protein (TP) level is a relatively unstable biochemical parameter that changes under the influence of external and internal conditions. TP levels, as an important index of innate immunity, can be good indicators of the health status of fish. Akbari et al. found increased TP levels in rainbow trout fed probiotics (72). In the present study, dietary supplementation of *E. asburiae* E7 had no significant impact on ACP activity in common carp, but this strain significantly upregulated plasma AKP activity and TP levels on day 21. The results suggest that strain E7 may not affect the host through nonspecific immunity *in vivo*.

During normal cellular respiration, SOD and GSH are the most important enzyme scavengers. GSH plays a key part in the regulation of oxidative stress in redox homeostasis by serving as an antioxidant and protecting cells by trapping free radicals (73). SODs, as the first barrier against oxidative stress, can eliminate excess reactive oxygen species (ROS) and prevent cellular damage caused by free radicals and is ubiquitous in every subcellular compartment (74). Measurement of T-AOC in serum or plasma can provide general information on the antioxidant status of the host (75). This study indicates that GSH activity was significantly downregulated within 21 days of feeding E7 compared with the control groups, while it was significantly upregulated at 28 days. In contrast, serum SOD activity was significantly lower than that of the control groups in the fourth weeks of feeding, but no significant differences in T-AOC were observed. According to previous research, an imbalance in the generation of active substances and the inability of the body to detoxify these active substances is known as oxidative stress (76). The above results show that, on the one hand, *E. asburiae* E7 feeding will not cause adverse reactions such as oxidative stress *in vivo*. On the other hand, there may be a negative regulatory relationship between GSH and SOD, as they were increased and decreased after 4 weeks, respectively, which together regulate the body in a redox equilibrium to maintain good health.

Cytokines secreted by immune cells or other cells trigger inflammation at infection sites and invoke the arrival of phagocytes to eliminate the invading pathogen. *IL-1β* plays a crucial role in the immune response in fish and mammals. In particular, the immune response is stimulated by inducing the release of cytokines that trigger macrophages, natural killer (NK) cells, and lymphocytes or by activating lymphocytes during tissue damage and pathogen invasion (77). As a kind of indispensable proinflammatory cytokine and primary neutrophil chemokine, *IL-8* is involved in immune cell trafficking that recruits and activates macrophages and neutrophils for killing microorganisms (78). *TNF-α* is a crucial proinflammatory cytokine that regulates immune function and mediates inflammatory responses, which can destroy various pathogens by stimulating different cellular responses, and, for this reason, it is considered an exceptional health indicator and biomarker for fish and mammals (79). In this study, a significant upregulation of *IL-1β*, *IL-8*, and *TNF-α* gene expression was observed in common carp kidneys after 28 days of *E. asburiae* E7 supplementation feeding. According to

previous studies, increased expression of proinflammatory cytokines after probiotics feeding is associated with enhanced innate immunity.

*IL-10* family cytokines play important roles in maintaining tissue homeostasis during infection and inflammation by suppressing excessive inflammatory responses, upregulating innate immunity, and promoting tissue repair mechanisms (80). The primary functions of *TGF-β*, maintaining immune tolerance and internal environmental homeostasis and regulating all aspects of the immune response, are essential for the development and maturation of immune cells. After acute tissue injury, *TGF-β* becomes a major regulator of the healing process, impacting all cell types involved (81). Similar to *IL-10*, *TGF-β* is considered an anti-inflammatory cytokine. The present study showed that *IL-10* mRNA expression continued to be upregulated throughout the 28-day feeding period, but *TGF-β* was not significantly changed overall, except for a significant increase at the third week. Therefore, we hypothesized that *E. asburiae* E7 did not cause damage to physiology because of no continuous upregulation in the repair factor *TGF-β*, and the upregulation of *IL-10* might be to balance the excessive inflammatory response caused by proinflammatory factors.

Lysozyme is a key defense molecule in the intrinsic immune system of fish and acts as a barrier to microbial invasion. Lysozyme kills sensitive bacteria by causing the breakdown of peptidoglycan in bacterial cell walls, resulting in rapid lysis of Gram-positive bacteria (70). The present results showed that the expression of the *lysozyme* gene was upregulated throughout the 28-day feeding period, demonstrating that the resistance of common carp against pathogenic bacteria likely is associated with the upregulated *lysozyme*. *IFN* is a subset of class II cytokines that plays a key role in host immune defense, especially in the immune response against viruses (82). Interestingly, *IFN* was upregulated after 28 days of feeding, indicating that E7 may have a potential antiviral effect, but this conjecture needs to be further verified.

Enhancing the disease resistance of aquatic animals is an important property to be considered for putative probiotic candidates. The present results showed that feeding E7-supplemented feed for 28 days significantly diminished the mortality of common carp challenged by *A. veronii*, reflecting an excellent protective effect *in vivo*. These data were similar to those of studies showing that diet supplementation with probiotics improved disease resistance in aquatic animals (83, 84). However, further studies on the optimal method of administration and probiotic dose are needed to avert probable adverse effects.

To summarize, our study was the first to evaluate the efficiency of *E. asburiae* as a prospective probiotic for aquaculture applications. We indicated that the resistance of common carp to *A. veronii* was enhanced by feeding a diet containing $1 \times 10^7$ CFU/g *E. asburiae* E7 for 28 days, but growth was not improved. Strain E7 can act as an immunostimulant to induce the upregulation of some innate cellular and humoral immune responses, resulting in enhanced resistance against *A. veronii*. Hence, the continuous activation of immune cells can be maintained by adding suitable fresh probiotics to the diet. E7 has the potential to act as a probiotic agent for green, sustainable aquaculture and aquatic product safety. Nevertheless, more research is needed to explore other modes of action of this probiotic, such as intestinal colonization, to further elucidate its efficacy and mechanisms.

## MATERIALS AND METHODS

**Probiotics isolation and identification.** Strain E7 used in the present research was isolated from the healthy common carp gut. Intestinal tissues from healthy common carp were collected under sterile conditions, rinsed with sterile saline (0.85%), and subsequently homogenized with sterile saline. The bacterial solution was diluted 1,000-fold with sterile saline. The diluted suspension (0.1 mL) was then spread on eight different kinds of medium agar plates, such as Luria-Bertani agar (LB), *Enterococcus* agar (EA), Pfizer *Enterococcus* selective agar (PSE), MRS agar (MRS), freshwater fish agar medium (FWA), *Bacillus megatherium* medium (BM), brain heart infusion agar medium (BHI), and tryptone soy agar (TSA), and cultivated at 28°C for 24 h. Different single colonies were picked randomly according to different phenotypes. The isolation bacterial sampling process was repeated to acquire pure colonies. These colonies were picked with a sterile inoculum needle and inoculated on LB plates coated with pathogenic indicator bacteria. Following incubation at 28°C for 24 h, antibacterial activity was shown in the inoculated

circles around the inoculation site, which was an important reference for selecting isolates. The strains with antimicrobial activity were preserved for further study.

Bacteria were identified by physiological and biochemical tests and Gram stained according to the manufacturer's instructions (Qingdao Hope Bio-Technology Co., Ltd., China). Subsequently, genomic DNA of candidate strains was extracted using a genomic DNA kit (Accurate Biotech [Hunan] Co., Ltd., China). The sequences of the 16S rRNA gene and *gyrB* were amplified by PCR using the following primers: 16S rRNA gene, 27F forward primer (5'-AGAGTTTGATCMTGGCTCAG-3') and 1492 R reverse primer (5'-TACGGTTACCTTGTTACGACTT-3'), and *gyrB*, forward primer (5'-GAAGTCATCATG GTTCTGCAYGCNGGGNAARTTYGA-3') and reverse primer (5'-AGCAGGGTACGGATGTGCGAGCCRTCNAC RTCNGCRTCNGTCAT-3'). The PCR conditions were as follows: 1 cycle at 95°C for 5 min, followed by 32 cycles at 95°C for 30 s, 55°C for 30 s, and 72°C for 60 s, and 1 cycle at 72°C for 10 min. The PCR products were sequenced (Sangon Biotech [Shanghai] Co., Ltd., China) to obtain nucleotide sequences and were subjected to similarity analysis and multiple sequence alignment by NCBI BLAST. Phylogenetic analysis of the 16S rRNA and *gyrB* gene sequences was performed for accurate identification of the strain. The purified strains identified were then stored in 50% (wt/vol) glycerol at $-80°C$ for subsequent studies.

**Antimicrobial assessment.** Sixteen common aquatic pathogenic bacteria (*A. hydrophila*, *A. media*, *A. veronii*, *A. jandaei*, *A. caviae*, *A. enteropelogenes*, *A. schubertii*, *A. salmonicida*, *Vibrio alginolyticus*, *V. cholerae*, *Pseudomonas aeruginosa*, *Ps. putida*, *Plesiomonas shigelloides*, *Acinetobacter johnsonii*, *Shewanella* sp., and *Citrobacter braakii*) and the strains from gut were incubated in LB broth medium at 28°C for 24 h at 160 rpm. Subsequently, 0.1 mL of pathogen culture diluted to a concentration of $1 \times 10^7$ CFU per mL by colony counting method was plated on LB agar plates. After all bacterial solutions were absorbed by agar plates, sterile paper discs of 6 mm in diameter containing 10 $\mu$L of the isolated suspension ($1 \times 10^7$ CFU/mL) were placed on the plate and coincubated at 28°C for 24 h. Antibacterial activity was detected by the transparent area around the inoculated paper. All aquatic pathogenic bacteria used in this study were supplied by The Fisheries Disease Lab of Northwest A&F University.

**Hemolytic test.** To determine hemolytic activity, isolates were inoculated on goat blood agar plates (Guangdong Huankai Biotechnology Co., Ltd.) to detect hemolysin. The activated strain E7 was punctured on a blood agar plate with an inoculating needle and incubated at 28°C for 24 h. Hemolysis was determined according to the following hemolysis types. Alpha-hemolysin production results in the decomposing red blood cells and a green ring formed around the colony. Beta-hemolysin forms a clear ring around the colony. No transparent loops or green areas were regarded as nonhemolytic. *A. veronii* was used as a positive control for hemolysis.

**Injection assay.** Twenty fish were equally divided into two groups, then each fish was injected intraperitoneally (i.p.) with 0.1 mL of strain E7 suspension ($1 \times 10^9$ CFU/mL) and sterile phosphate-buffered saline (PBS; 0.1 M, pH 7.4) to observe the survival rate within 7 days.

**Antibiotic sensitivity assay.** The sensitivity of E7 strains to antibiotics was evaluated by the paper diffusion method. Strain E7 (0.1 mL; $1 \times 10^7$ CFU/mL) was coated on LB agar plates, and discs containing 35 medical antibiotics (Hangzhou Binhe Microorganism Reagent Co., Ltd.) and 11 aquatic/veterinary antibiotics were placed on the surface of LB agar and incubated for 24 h at 28°C. The diameters of zones of inhibition were measured, and the antimicrobial susceptibility was divided into susceptible (S), resistant (R), and intermediate resistant (IR) according to the diameter of inhibition circle.

**Antibiotic resistance gene (ARG) detection.** PCR was used to determine whether the bacteria carried ARGs. The quality and concentration of the amplified products were evaluated by agarose gel electrophoresis (1.5% agarose in 1$\times$ Tris-acetate-EDTA [TAE] buffer) and UV. Fourteen common ARG types in water environments were selected, including three sulfonamide resistance genes (*sul1*, *sul2*, and *sul3*), three tetracycline resistance genes (*tetA*, *tetC*, and *tetM*), two quinolones resistance genes (*qnrA* and *qnrS*), two chloramphenicol resistance genes (*cmlA* and *floR*), two $\beta$-lactam resistance genes (*blaTEM* and *blaCMY*), one macrolide resistance gene (*ermA*), and one aminoglycoside resistance gene (*aac[6']-Ib*) (Table S1 in the supplemental material). The total PCR volume was 20 $\mu$L, consisting of 10 $\mu$L of 2$\times$ *Taq* master mix, 8 $\mu$L of double-distilled water (ddH$_2$O), 10 pmol/L forward and reverse primers, and 1.0 $\mu$L of bacterial genomic DNA. After initial enzyme activation at 94°C for 10 min, 32 cycles of amplification (denaturation at 94°C for 30 s, annealing at 60°C for 30 s, and extension at 72°C for 10 s) were performed, followed by extension at 72°C for 10 min. The PCR product was analyzed by electrophoresis in a 1.5% agarose gel for 40 min at 100 V and 50 mA.

**pH, temperature, and bile salt tolerance assessment.** Tolerance to pH, temperature, and bile salts was assessed for E7 based on previous studies, with some modifications (85). LB broth medium with different pH values (1, 2, 3, 4, 5, 6, and 7) was adjusted with HCl (1 mol/L), and LB broth medium containing different concentrations of bile salts (0 [control], 0.1, 0.2, 0.3, 0.4, 0.5, 0.6, 0.7, 0.8, 0.9, 1, 2, 3, and 4% [wt/vol]) were prepared. The activated E7 ($1 \times 10^7$ CFU/mL) was inoculated at 1% (vol/vol) in LB broth medium with different pH values and bile salt concentrations, which were incubated at 28°C and 160 rpm for 24 h. Activated strain E7 was inoculated at 1% (vol/vol) into LB broth medium and incubated at 35, 40, 45, 50, 55, 60, 65, and 70°C for 24 h in a thermostat water bath to determine temperature tolerance. The absorbance at a wavelength of 600 nm was measured using 0.2 mL of the above bacterial culture. Meanwhile, 0.1 mL of bacterial culture was coated on LB solid medium and cultivated at 28°C for 24 h. Photographs were taken, and bacterial tolerance was recorded.

**Experimental design.** *E. asburiae* E7 was obtained using the above isolation methods and was determined to be nonpathogenic by *in vitro* hemolysis assay and *in vivo* injection assay and could be used for subsequent diet supplementation. The E7 isolate was inoculated onto LB agar plates using bacterial inoculators for pure culture, and single colonies were inoculated into LB broth medium for

**TABLE 4** Primer sequences for RT-qPCR

| Gene name | Sequences of primers (5′ to 3′)[a] | Accession no. |
|---|---|---|
| *β-Actin* | F: AGACATCAGGGTGTCATGGTTGGT | M24113.1 |
| | R: CTCAAACATGATCTGTGTCAT | |
| *IL-1β* | F: ACCAGCTGGATTTGTCAGAAG | AB010701.1 |
| | R: ACATACTGAATTGAACTTTG | |
| *IL-10* | F: TGATGACATGGAACCATTACTGG | AB110780 |
| | R: CACCTTTTTCCTTCATCTTTTCA | |
| *TNF-α* | F: GGTGATGGTGTCGAGGAGGAA | AJ311800.1 |
| | R: TGGAAAGACACCTGGCTGTA | |
| *TGF-β* | F: ACGCTTTATTCCCAACCAAA | DQ411314.1 |
| | R: GAAATCCTTGCTCTGCCTCA | |
| *IFN* | F: GTCGCTGCTGCTTGATAGAA | AM261214 |
| | R: CTGAAGCTCCCTCCATACTT | |
| *IL-8* | F: GTCTTAGAGGACTGGGTGTA | AB470924.1 |
| | R: ACAGTGTGAGCTTGGAGGGA | |
| *Lysozyme* | F: GTGTCTGATGTGGCTGTGCT | AB027305.1 |
| | R: TTCCCCAGGTATCCCATGAT | |

[a]F, forward; R, reverse.

proliferation at 28°C for 24 h. The concentration was calculated by the colony counting method after 10-fold gradient dilution of the E7 culture suspension with sterile PBS.

Common carp (*Cyprinus carpio*) weighing $0.57 \pm 0.06$ g were acquired from Guangzhou Menghu fish seed farm (Guangdong, China). Fish were housed in tanks (400-L water capacity) at 23 to 25°C for 30 days before the start of the experiment and were adapted to laboratory conditions (dissolved oxygen: $5.50 \pm 0.82$ mg/L; pH: $7.54 \pm 0.45$; nitrites: $0.012 \pm 0.004$ mg/L; ammonia: $0.130 \pm 0.024$ mg/L). Fish were fed at 2% of body weight twice every day (8:30 and 16:30). The water in the experimental tank was replaced one-third of the way through every day. The water used in the study was aerated 24 h in advance and adjusted to the optimum temperature.

The method for producing diets containing live bacteria is based on Chi et al. (86), as described below. A known concentration of the *E. asburiae* E7 suspension was slowly sprayed into the feed and mixed well. The number of live bacteria in the diet was determined by spreading onto LB plates to ensure that the diet contained $10^7$ CFU/g E7. Commercial feed sprayed with sterile PBS was used as a control. The experimental diets were lyophilized and ground into pellets of proper sizes and stored at −20°C. Fresh diets were made every 7 days to ensure that there was an adequate number of live bacteria in the experimental feed.

Three hundred fish of similar sizes were randomly divided into two groups ($n = 150$), with 3 replicates of 50 fish in each group. Different diets were fed to each group: three tanks were added without the base diet (control group), and the other three tanks were provided with E7 at $1 \times 10^7$ CFU/g of the strain diet. Each tank represented one biological repetition per group.

**Sample collection.** After 7, 14, 21, and 28 days of rearing, 6 fish were stochastically selected from each replicate, and blood samples were collected. Following euthanasia of the fish with MS-222, blood samples were collected from the broken neck using a pipette, loaded into sterile Eppendorf tubes, and left to stand for 6 h at 4°C. After centrifugation at 4°C and 3,000 rpm for 5 min, serum was obtained and stored at −80°C. Moreover, fish kidney tissue was sampled and soaked in 0.2 mL of TRIzol reagent (TaKaRa, Japan) for subsequent RNA extraction.

**Growth performance measurement.** To reflect the macroscopic effect of E7 on common carp growth, 15 fish were randomly selected from each replicate every 7 days to determine body weight and length, and 6 were picked to measure carcass weight (excluding viscera weight).

**Nonspecific immunological parameter determination.** Serum was obtained once every 7 days to measure acid phosphatase (ACP) activity, alkaline phosphatase (AKP) activity, total protein level (TP), glutathione level (GSH), total antioxidant capacity (T-AOC), and superoxide dismutase (SOD) activity. Measurements were performed using commercial kits from Nanjing Jiancheng Institute following the manufacturer's instructions.

**RNA extraction and real-time quantitative PCR (RT-qPCR).** The detailed procedures for total kidney RNA extraction and RT-qPCR are as performed by Li et al. (87). Primers are listed in Table 4, and the *β-actin* gene was used as a reference gene. The PCR cycling conditions were:1 cycle at 95°C for 180 s and 40 cycles at 95°C for 20 s, 60°C for 30 s, and 72°C for 20 s. Melting curves and analyses were performed as described in Li et al. (87).

**Histopathological analysis.** After 28 days of E7 feeding, the liver, spleen, kidney, and intestine of fish were fixed with 4% paraformaldehyde. The tissues were paraffin embedded and processed according to routine histological procedures. Histological sections (3 $\mu$m) were stained with hematoxylin and eosin (H&E) and examined using a light microscope equipped with an imaging system (Olympus BX53).

**Challenge test.** Pretests were performed before the challenge. After incubation of *A. veronii* in LB broth for 24 h at 28°C, the culture was diluted to obtain $1 \times 10^5$, $1 \times 10^6$, $1 \times 10^7$, $1 \times 10^8$, and $1 \times 10^9$ CFU/mL suspensions. Every 10 common carp were i.p. injected with 0.1 mL of the above-mentioned bacterial suspension with different dilution concentrations, and the results revealed that the 7-day 50%

lethal dose ($LD_{50}$) of *A. veronii* was 9.857 $\times$ $10^6$ CFU/mL (Fig. S2); thus, the concentration used for the challenge test was approximately 1 $\times$ $10^7$ CFU/mL. A formal test was then performed. After 28 days of feeding, 0.1 mL of a suspension of *A. veronii* (1 $\times$ $10^7$ CFU/mL) was injected i.p. into each fish from experimental and control groups ($n = 50$). Survival rate (number of survivors/total number) $\times$ 100 was used to access the protective efficacy of the E7 strain against *A. veronii*. Pathological changes were observed daily for 8 days, and the dead fish were removed promptly.

**Statistical analysis.** Statistical analyses and drawing of all data were performed using GraphPad Prism 8.0 (GraphPad Software, Inc., San Diego, CA, USA). Normally and nonnormally distributed data were compared using independent-samples *t* tests or a Mann-Whitney *U* test, respectively. All results are presented as the mean $\pm$ standard deviation (SD). Single asterisks (*, $P < 0.05$) and double asterisks (**, $P < 0.01$) indicate statistical significance compared with control. A *P* value of $<0.05$ and a *P* value of $<0.01$ were regarded as statistically significant and extremely significant, respectively.

**Data availability.** The 16S rRNA and *gyrB* sequences of E7 are available in public databases with accession numbers OP522325 and OP558470.

## SUPPLEMENTAL MATERIAL

Supplemental material is available online only.

**SUPPLEMENTAL FILE 1**, PDF file, 0.4 MB.

## ACKNOWLEDGMENTS

Financial support for this study was provided by the Key Research and Development Projects of Shaanxi Province under grant 2021NY-025. National Key Research and Development Program of China (2022YFD2401200).

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
