## [Reviewer comments · Microbiology Spectrum]

Microbiology Spectrum

***Enterobacter asburiae* E7, a novel potential probiotic, enhances resistance to *Aeromonas veronii* infection via stimulating the immune response in common carp (*Cyprinus carpio*)**

Jing Li, Zhao Zhang, Zhi-Bin Wu, Shen-Ye Qu, Gao-Xue Wang, and Fei Ling

Corresponding Author(s): Fei Ling, Northwest A&F University College of Animal Science and Technology

Review Timeline:

Submission Date:	October 19, 2022
Editorial Decision:	December 17, 2022
Revision Received:	February 16, 2023
Accepted:	February 24, 2023

Editor: Jianjun Wang

Reviewer(s): Disclosure of reviewer identity is with reference to reviewer comments included in decision letter(s). The following individuals involved in review of your submission have agreed to reveal their identity: Xiaoming Wang (Reviewer #2)

Transaction Report:

DOI: <https://doi.org/10.1128/spectrum.04273-22>

Prof. Fei Ling
College of Animal Science and Technology, Northwest A&F University
Yangling
China

Re: Spectrum04273-22 (*Enterobacter asburiae* E7, a novel potential probiotic, enhances resistance to *Aeromonas veronii* infection via stimulating the immune response in common carp (*Cyprinus carpio*))

Dear Prof. Fei Ling:

I have received the reviews of your manuscript entitled "*Enterobacter asburiae* E7, a novel potential probiotic, enhances resistance to *Aeromonas veronii* infection via stimulating the immune response in common carp (*Cyprinus carpio*)", and I regret to inform you that we will not be able to publish it in Spectrum. Your submission was read by reviewers with expertise in the area addressed in your study and it was the consensus view of these reviewers that your paper did not meet the standards necessary for publication. Copies of the reviewers' comments are attached for your consideration.

I am sorry to convey a negative decision on this occasion, but I hope that the enclosed reviews are useful. Please note, rejections from Microbiology Spectrum are final and your manuscript will not be considered by other ASM journals. We wish you well in publishing this report in another journal and hope that you will consider Spectrum in the future.

Sincerely,

jianjun wang
Editor, Microbiology Spectrum

Reviewer comments:

Reviewer #1 (Comments for the Author):

After thoroughly reviewing the manuscript (Spectrum04273-22), I suggest that this manuscript should have a minor revision. The work of Jing Li et al. studied a novel potential probiotic *Enterobacter asburiae* E7 enhances resistance to *Aeromonas veronii* infection via stimulating the immune response in common carp. In general, the research described in this manuscript is quite interesting and of great importance for our understanding of the functional role of *E. asburiae* as feed supplementary, this study is of great value to develop ecologically efficient and environmentally friendly method to control disease. *E. asburiae* strain E7 was isolated from the intestinal tract of healthy common carp, and its probiotic potential was systematically evaluated in vitro and in vivo. E7 was antagonistic to major *Aeromonas* pathogens, and dietary supplementation enhanced the resistance to *A. veronii* via stimulating the immune response in common carp. Overall, the study is well carried out and gets valuable conclusions. However, there are several important issues that should be addressed before publication. I have the following overall suggestions:

1. The "INTRODUCTION" is lengthy and it is recommended to refine the language.
2. Please add a subheading.
3. Line 61: "simpler" should be changed to "simple".
4. Line 63: "already poses" should be changed to "already have posed".
5. Line 95: "*Aeromonas*" is not a specific disease name.
6. Line 205: "($P < 0.05$, Fig. 7C)" should be "($P < 0.05$, Fig. 7C)".
7. It is very hard to take blood from the small fish via caudal vein especially in cyprinids. Authors should explain how much volume blood they took in a fish.
8. Figure 1: The last picture should be "*Shewanella* sp."
9. The legend of Figure 4: It is suggested explain the abbreviation "ip" at the end of the note.
10. The legend of Figure 6: What does "n=45/n=18" mean? Why are there two quantities?
11. Table 2 shows the total size of the inhibition zone. It is suggested to add the original diameter of the paper discs in the table note to avoid confusion.

Reviewer #2 (Comments for the Author):

Probiotics can be an alternative strategy to antibiotics , and this paper described *E. asburiae* E7 is a promising new Gram-negative probiotic that can enhance the health condition and bacterial resistance of aquatic animals. But I still have some questions remaining to be clarified.

- 1)The English writing of this manuscript should be checked and improved by a native English speaker.
- 2)Based on Figure 1, authors stated that *E. asburiae* E7 against common aquatic pathogenic bacteria, So what are the criteria for *E. asburiae* E7 to be antibacterial? You can't define that *E. asburiae* E7 has antibacterial effect just according to a little bit of a bacteriostatic zone emergence.
- 3)Safety evaluation need more rigorous, for example, Whether the E7 strain causes damage to cells and animal tissues should be taken into account.
- 4)In this paper, authors concluded that *E. asburiae* E7 is a promising new Gram-negative probiotic. It should be added which resistance genes the strain carries and whether it will develop resistance to antibiotics commonly used in fish.
- 5)The picture is too fuzzy.
- 6)Pay attention to the reference case, italics and other problems, according to the requirements of the magazine.

Dear editors and reviewers:

On behalf of my co-authors, we thank you very much for giving us an opportunity to revise our manuscript, and also for the reviewers' comments concerning our manuscript entitled "*Enterobacter asburiae* E7, a novel potential probiotic, enhances resistance to *Aeromonas veronii* infection via stimulating the immune response in common carp (*Cyprinus carpio*)". Those comments are all valuable and very helpful for revising and improving our paper, as well as the important guiding significance to our study. We have studied comments carefully and have made correction which we hope meet with approval. Moreover, we carefully checked the tense and grammar of each sentence and corrected the mistakes in this manuscript. The corrected content was highlighted in red color. The main corrections in the paper and the responses to the reviewer's comments are as follows:

Response to reviewer #1:

1. The "INTRODUCTION" is lengthy and it is recommended to refine the language.

Thanks for the reviewer's valuable and professional suggestion. We have refined the content of INTRODUCTION.

2. Please add a subheading.

Thanks for the reviewer's advice. We have added a running title called "*Enterobacter asburiae* E7 enhances host immunity".

3. Line 61: "simpler" should be changed to "simple".
4. Line 63: "already poses" should be changed to "already have posed".
5. Line 95: "*Aeromonas*" is not a specific disease name.
6. Line 205: " $(P < 0.05, \text{Fig. 7C})$ " should be " $(P < 0.05, \text{Fig. 7C})$ ".

Thanks for the reviewer's careful examination. These mistakes were carefully corrected in the manuscript.

7. It is very hard to take blood from the small fish via caudal vein especially in cyprinids.

Authors should explain how much volume blood they took in a fish.

Thanks for the reviewer's professional and careful review. We collected blood samples from the broken neck using pipette, not from the tail vein. The blood of each fish was about 20 μL . This method was described on line 489.

8. Figure 1: The last picture should be "*Shewanella* sp."

9. The legend of Figure 4: It is suggested explain the abbreviation “ip” at the end of the note.

Thanks for the reviewer's comment. We have modified Figure 1 and supplemented the full name of “ip” for Figure 4.

10. The legend of Figure 6: What does “n=45/n=18” mean? Why are there two quantities?

Thanks for the reviewer's professional and careful review. 45 fish in each group were randomly selected to measure body length and body weight, and 18 of them were measured carcass weight. In order for the test to proceed smoothly and to ensure sufficient numbers of fish, a smaller number of carcass weight are measured. We have added the meaning of “n=45/n=18” in Figure 6 legend.

11. Table 2 shows the total size of the inhibition zone. It is suggested to add the original diameter of the paper discs in the table note to avoid confusion.

Many thanks for the comments. We have supplemented the original diameter of the paper discs in the table note.

Response to reviewer #2:

1. The English writing of this manuscript should be checked and improved by a native English speaker.

Thanks for the reviewer's comments. This manuscript has been checked and improved by native English speaker.

2. Based on Figure 1, authors stated that *E. asburiae* E7 against common aquatic pathogenic bacteria, So what are the criteria for *E. asburiae* E7 to be antibacterial? You can't define that *E. asburiae* E7 has antibacterial effect just according to a little bit of a bacteriostatic zone emergence.

Thanks for the reviewer's professional and careful review. Screening criteria for probiotics is based on antibacterial activity. The antibacterial effect of probiotics can be directly judged by examining the presence or size of the inhibition zones of probiotics against pathological bacteria. A large number of studies have used the method of antagonistic experiment to define antibacterial activity, so it is reasonable to define the antibacterial effect of E7 according to the inhibition zones. The following are early and recent references to this method:

[1] O'Sullivan DJ. 2001. Screening of intestinal microflora for effective probiotic bacteria. J Agric

Food Chem 49:1751-1760.

[2] Schillinger U, FK Lücke. 1989. Antibacterial activity of *Lactobacillus sake* isolated from meat. Appl Environ Microbiol 55:1901-1906.

[3] Fleming HP, Etechells JL, Costilow RN. 1975. Microbial inhibition by an isolate of *Pediococcus* from cucumber brines. Appl Environ Microbiol 30:1040-1042.

[4] Leska A, Nowak A, Szulc J, Motyl I, Czarnecka-Chrebelska KH. 2022. Antagonistic Activity of Potentially Probiotic Lactic Acid Bacteria against Honeybee (*Apis mellifera* L.) Pathogens. Pathogens 11:1367.

3. Safety evaluation need more rigorous, for example, Whether the E7 strain causes damage to cells and animal tissues should be taken into account.

Thanks for the reviewer's valuable and professional suggestion. In the experiment, histopathological sections of liver, spleen, kidney and intestine of common carp after feeding E7 for 28 days were completed and observed, and no pathological changes were found. Due to the large number of pictures in this paper, we did not add histopathological pictures, and only showed the results of no hemolytic and no pathogenicity to the host, which proved that E7 was safe. We think the reviewer's consideration is reasonable, so we added the histopathological pictures in Figure 4C.

4. In this paper, authors concluded that *E. asburiae* E7 is a promising new Gram-negative probiotic. It should be added which resistance genes the strain carries and whether it will develop resistance to antibiotics commonly used in fish.

Thanks for the reviewer's professional comments. We accepted the reviewer's valuable suggestions, so we supplemented some tests. We examined whether E7 carries resistance genes and its sensitivity to common aquatic / veterinary antibiotics. These results are shown in Supplementary Fig. 3 and Table 4. Meanwhile, we have supplemented and modified corresponding descriptions in the materials and methods, results and discussion sections of the manuscript on line 142-159, line 153-155, line 158-159, line 262-264, line 269-275, line 430, line 434-446, line 501-505.

5. The picture is too fuzzy.

Many thanks for the comments. We will upload a clear picture with high resolution when we submit again.

6. Pay attention to the reference case, italics and other problems, according to the requirements of the magazine.

Thanks for the reviewer's professional and careful review. We have carefully revised the references according to the format of the journal.

February 24, 2023

Prof. Fei Ling
Northwest A&F University College of Animal Science and Technology
Yangling
China

Re: Spectrum04273-22R1-A (*Enterobacter asburiae* E7, a novel potential probiotic, enhances resistance to *Aeromonas veronii* infection via stimulating the immune response in common carp (*Cyprinus carpio*))

Dear Prof. Fei Ling:

Your manuscript has been accepted, and I am forwarding it to the ASM Journals Department for publication. You will be notified when your proofs are ready to be viewed.

Sincerely,

Jianjun Wang
Editor, Microbiology Spectrum
